# Effects of Different Drying Methods on the Selenium Bioaccessibility and Antioxidant Activity of *Cardamine violifolia*

**DOI:** 10.3390/foods12040758

**Published:** 2023-02-09

**Authors:** Peiyu Wang, Yue Li, Ruipeng Yu, Dejian Huang, Shangwei Chen, Song Zhu

**Affiliations:** 1State Key Laboratory of Food Science and Technology, Jiangnan University, Wuxi 214122, China; 2International Joint Laboratory on Food Safety, Jiangnan University, Wuxi 214122, China; 3School of Food Science and Technology, Jiangnan University, Wuxi 214122, China; 4Department of Food Science and Technology, National University of Singapore, Singapore 117543, Singapore

**Keywords:** selenium, Se species, *Cardamine violifolia*, drying methods, bioaccessibility, antioxidant

## Abstract

Understanding the effects of drying on the selenium (Se) content and Se bioaccessibility of Se-rich plants is critical to dietary supplementation of Se. The effects of five common drying methods (far-infrared drying (FIRD), vacuum drying (VD), microwave vacuum drying (MVD), hot air drying (HD), and freeze vacuum drying (FD)) on the content and bioaccessibility of Se and Se species in *Cardamine violifolia* leaves (CVLs) were studied. The content of SeCys_2_ in fresh CVLs was the highest (5060.50 μg/g of dry weight (DW)); after FIRD, it had the lowest selenium loss, with a loss rate of less than 19%. Among all of the drying processes, FD and VD samples had the lowest Se retention and bioaccessibility. FIRD, VD, and FD samples have similar effects on antioxidant activity.

## 1. Introduction

Selenium (Se) is an essential trace element that was first known for its toxic effect. Later, in 1957, it was shown that Se is more effective than vitamin E and cystine in preventing necrotizing liver disease [1], demonstrating its nutritional benefit. The scientific community has paid increasing attention to the link between Se and human health in the decades since then. In humans, the cellular and organismal functions of Se are mediated by 25 selenoproteins whose active center is selenocysteine, which is encoded by the UGA codon [2]. Selenoproteins are essential for immunological, cognitive, and reproductive health. Inadequate Se intake in the diet causes certain disorders, including Keshan disease and Kashin-Beck disease, as well as an increase in the incidence of cancer [3]. Se is abundant in nature and can be found in both inorganic and organic forms in food. Different Se species exhibit markedly different physicochemical and biological properties [4], with organic Se being less toxic and safer than inorganic Se. Organic Se species (SeCys_2_, SeMet and MeSeCys) have been reported to be three times more effective than inorganic Se (Se(VI) and Se(IV)) in alleviating symptoms of deficiency [5].

As a result, Se dietary supplementation should take into account the variation across Se species while also considering the total Se content. *Cardamine violifolia* (*C. violifolia*) was the first discovered Se hyperaccumulator from the genus Cardamine with unique properties in terms of Se accumulation [6], which is abundant in Enshi, a Se-enriched area of China. *C. violifolia* has a superb capacity for selenium accumulation and organic selenium conversion [7]. *C. violifolia* has a good antioxidant capacity which was not affected by the application of different exogenous selenium sources to *C. violifolia* [6]. However, there is limited research on the bioaccessibility of Se and organic Se in *C. violifolia*. Se dietary supplementation should take into account the variation across Se species while also considering the total Se content. Se and Se species in plants are less stable, particularly after processing, resulting in considerable Se losses [8,9]. The total Se content of potatoes decreased by 31.7% after frying; rice grain lost 5.6%; and *Pleurotus eryngii* lost 18.0–35.8% [10]. Boiling causes more Se loss than frying does. In terms of Se species loss, the study discovered that SeMet, MeSeCys, and SeCys_2_ lost 19.7%, 35.4%, and 16.4% after boiling, respectively, compared to 40%, 16.1%, and 53.6% after frying. The loss rate of the dominant species SeMet after boiling corn grains is higher than after frying them [10]. Furthermore, in cooked grains, SeCys_2_ and MeSeCys are completely lost. This shows that the loss law of Se species varies depending on the samples, which could be connected to the plant structure and dominant species.

The oral bioaccessibility of a drug is the portion of a substance that is soluble in the gastrointestinal environment and available for absorption, according to the definition published in 1999. Determination of the bioaccessibility depends on the relationship between the results of a specific in vitro testing system and an appropriate in vivo model [11]. Therefore, Se bioaccessibility as a professional term is used to describe the portion that may be absorbed by the gastrointestinal tract after oral supplementation of Se, which is significantly different from the actual absorption in the body. However, in vitro simulation is also a widely used evaluation method with the advantages of simplicity, short time, and nonspecificity compared to animal experiments [12]. To date, the Se bioaccessibility of *C. violifolia* has not been reported, and its bioaccessibility data could be an important guide for future dietary Se supplementation application.

The bioaccessibility of Se in Se-rich foodstuffs has been studied in cereals, vegetables, chlorella, yeast, and fish [13]. The bioaccessibility of Se decreases after the soaking and fermentation of related cereals (chickpeas, etc.), and the bioaccessibility (14.3 ± 1.2%) of chick peas is the highest [14]. After boiling and frying potatoes, the bioaccessibility of total Se reached 64.7% and 23.4%, and the bioaccessibility of SeMet and SeCys_2_ after boiling were 18.5% and 68.5%, respectively, but MeSeCys was not bioaccessible [4]. To date, there has been no research regarding drying influences on Se bioaccessibility, and no report on *C. violifolia* has been published.

The purpose of the present study was to compare the effects of far-infrared drying (FIRD), vacuum drying (VD), microwave vacuum drying (MVD), hot air drying (HD), and freeze vacuum drying (FD) on total Se, Se species, and Se bioaccessibility in CVLs. Different drying methods have their own characteristics. In the process of FIRD, electromagnetic wave energy is directly absorbed by the material, VD is suitable for substances with low vacuum or easy oxygen reaction, MVD can overcome the disadvantage of slow heat conduction of VD, HD has simple principle characteristics and can be used for drying many substances, and FD has better retention, which is useful for the quality and nutrition of materials. Therefore, five commonly used drying methods were selected for experimental study. In addition, the physicochemical properties (microstructure, color, total polyphenols, and flavonoids) and antioxidant activity of the dried CVLs were analyzed in this study.

## 2. Materials and Methods

### 2.1. Materials

*C. violifolia* leaves (CVLs) were supplied by Hubei Enshi Deyuan Health Technology Development Co., Ltd. (Enshi, Hubei, China). Pancreatin from porcine pancreas (P7545, 8 × USP), pepsin (P7000, ≥250 U/mg), 2,2-diphenyl-1-picrylhydrazyl radical (DPPH), and 2,2′-azino-bis (3-ethylbenzothiazoline-6-sulfonic acid) diammonium salt (ABTS) were all purchased from Sigma–Aldrich (St. Louis, MO, USA). Trypsin and protease K were sourced from the Hefei Bomei Biotechnology Co., Ltd. (Hefei, China). Protease E was purchased from the Shanghai Yingxin Laboratory Equipment Co., Ltd. (Shanghai, China). Pork liver (GBW10051), Se(IV) (GBW10032), Se(VI) (GBW10033), MeSeCys (GBW10088), SeMet (GBW10034), and SeCys_2_ (GBW10087) were sourced from the National Research Center for Certified Reference Materials (Beijing, China). Trolox ((±)-6-hydroxy-2,5,7,8 tetramethylchromane-2-carboxylic acid) was purchased from the Shanghai Yuanye Bio-Technology Co., Ltd. (Shanghai, China). All reagents used were of analytical grade.

### 2.2. Drying Methods

The effects of different drying methods on the Se bioaccessibility of C. violifolia leaves (CVLs) was studied in this work. In this study, five common drying methods were selected to process CVLs, namely FIRD, VD, MVD, HD, and FD.

#### 2.2.1. Far-Infrared Drying (FIRD)

A laboratory-scale far-infrared dryer (Shanghai Reli Technology Co., Ltd. Shanghai, China) was utilized to process 150 g CLVs in this experiment. The temperature inside the FIRD oven was set to 60 °C by using the intelligent temperature controller, and automatic thermostat control was realized by matching the thermal element. The strong air pressure supply through the air duct allowed the internal temperature of the oven to achieve uniformity. The wavelength of infrared radiation under the working condition of the far-infrared heating light source was 5–15 μm.The drying processes were conducted at 60 °C for 7 h. The final moisture content of the CVLs was 0.041 ± 0.001 kg water/kg dry weight (DW).

#### 2.2.2. Vacuum Drying (VD)

Vacuum drying was carried out in a VD oven (DP33C, Yamao, Japan). The dryer was preheated to the desired temperature and vacuum level before the experiment. The degree of vacuum and the temperature were set at −0.098 MPa and 60 °C, respectively, for 11 h to obtain dried sheets of 0.046 ± 0.001 kg water/kg dry weight (DW).

#### 2.2.3. Microwave Vacuum Drying (MVD)

A laboratory-grade microwave vacuum dryer (Promis Tech, Wroclaw, Poland) was used for drying, and the temperature and vacuum were set at 60 °C, 800 W, and −0.098 MPa for 7 h. The final moisture content of the CVLs was 0.036 ± 0.001 kg water/kg dry weight (DW).

#### 2.2.4. Hot Air Drying (HD)

HD was performed in an electric drying oven (DHG-9053A, Yiheng, Shanghai, China) at 60 °C and 1 m/s air velocity for 4 h. After the drying process, the final moisture content of the CVLs was 0.069 ± 0.001 kg water/kg dry weight (DW).

#### 2.2.5. Freeze Drying (FD)

In FD, fresh CVLs were initially frozen at −50 °C in a cold trap for 30 min. The FD process of the CVLs was carried out in a freeze dryer (SCIENTZ-10N, Xinzhi, Ningbo, China). The vacuum pressure was set at 0.010 MPa, and the temperature of the freeze dryer cold trap was −60 °C. The drying process lasted 48 h to obtain dried sheets of 0.056 ± 0.001 kg water/kg dry weight (DW).

### 2.3. Determination of the Se Content

The appropriate sample mass was weighed and placed into Teflon digestion vessels, to which 3.5 mL of HNO_3_ was added. After overnight incubation, 5 mL of Milli-Q water was added. The microwave digestion instrument (MARS 6, CEM, Matthews, NC, USA) was performed for sample digestion. The program was as follows: heating at 10 min to 120 °C and keeping at 120 °C for 10 min; heating at 5 min to 150 °C and keeping for 10 min; heating at 5 min to 190 °C and keeping for 20 min; and cooling for 30 min. After cooling, the sample was fixed with Milli-Q water to 100ml and filtered to be tested. An inductively coupled plasma mass spectrometer (ICP–MS) (iCAP TQ, Thermo Fisher Scientific Inc., Waltham, MA, USA) was used in the analysis. When testing, the TQ-O_2_ reaction mode was used, the instrument monitored the ^80^Se isotope, and the sampling cone was a nickel cone. ^89^Y and ^45^Sc were used as the internal standard elements, and the control (pig liver GBW10051) was mass calibrated. All samples were measured three times in duplicate [15].

### 2.4. Determination of Se Species

Selenoderivatives attach to proteins primarily in plants, but selenoproteins are difficult to successfully obtain via acid or water extraction [4]. Protease K, trypsin, and alkaline protease were applied to extract bound Se from CVLs in order to attain a greater extraction rate [15]. First, dried CVL powder samples of 100 mg were weighed in the extraction bottle, 15 mL of Tris-HCl (pH = 8.5) solution was added, and the extract was agitated at 45 °C and 400 rpm for 4 h. Then, the sample was centrifuged at 4000 rpm for 30 min, and the supernatant was collected and stored at −20 °C after passing through a 0.45 μm membrane for further analysis.

The Se speciation after enzyme extraction and gastrointestinal digestion was determined by the optimized RPIC/ICP–MS method [15]. The system consisted of a coupled ICP–MS and HPLC (U3000, Thermo Fisher Scientific Inc., Waltham, MA, USA) equipped with a TechMateC18-ST RP (4.6 mm × 250 mm, 5 μm, 120 Å; SHISEIDO, Tokyo, Japan). The injection volume was 10 μL, separation was performed at 30 °C, the eluent was 30 mM diammonium hydrogen phosphate−0.05 mM tetrabutylammonium hydroxide−3% (*v*/*v*) methanol, the pH was adjusted to 6.0 with 10% (*v*/*v*) formic acid, and the flow rate was 1.0 mL/min. The ICP–MS determination model was the TQ-O_2_ model, and the ^80^Se isotope was determined. Five Se species (SeCys_2_, SeMet, MeSeCys, Se(IV), and Se(VI)) mixed standard solutions (100 μg Se/L) were prepared in Milli-Q water. By matching the retention times of the sample peak and the mixed standard peak, the concentration of the Se species was determined.

### 2.5. In Vitro Gastrointestinal Simulated Digestion Test

The bioaccessibility of Se and Se species in CVLs obtained by different drying methods was determined by gastrointestinal simulated digestion, referring to the method of Minekus et al. with slight modifications [16]. The gastrointestinal simulation experiment was divided into two stages: gastric digestion (G) and gastrointestinal digestion (GI).

A 100 mg powder sample was placed in a 20 mL digestion flask. Then, 3 mL of pepsin solution (7 mg/mL, enzyme activity 250 U/mg, prepared with 0.5 mol/L hydrochloric acid-potassium chloride solution at a pH of 1.5) was reacted with magnetic stirring at 37 °C for 2 h. Afterward, 2 mL of trypsin solution (8 mg/mL, enzyme activity 8 × USP, prepared with 0.5 mol/L acetic acid–sodium acetate buffer solution at a pH of 5.2) was added to the gastric food chyme and reacted with magnetic stirring at 37 °C for 2 h. Centrifugation at 4000 rpm for 30 min followed the inactivation of the enzyme. The total Se and Se species in the supernatant were investigated and analyzed after passing through a 0.45 μm membrane.

The bioaccessibility of Se and Se species in CVLs was expressed as BAC %, and the calculation formula was as follows [4]:(1)$$\begin{array}{c} \mathrm{BAC}\;\%=\frac{\mathrm{Se}\;\mathrm{or}\;\mathrm{Se}\;\mathrm{species}\;\mathrm{in}\;G\;\mathrm{or}\;\mathrm{GI}}{\mathrm{Se}\;\mathrm{or}\;\mathrm{Se}\;\mathrm{species}\;\mathrm{in}\;\mathrm{sample}}\times 100, \end{array}$$
where Se or Se species in G or GI are the total Se concentration or Se species concentration (μg/g) in G or GI digests, and Se or Se species in the sample are the total Se concentration or Se species concentration (μg/g) in the corresponding CVLs.

### 2.6. Scanning Electron Microscopy (SEM)

The dried CVLs were characterized by SEM, and the effect of different drying methods on the microstructure of CVLs was studied. The dried CLVs were sectioned, fixed on a filming table, and observed after gold spraying on the sample surface. Observation was carried out under SEM (SU8100, Hitachi Ltd., Ibaraki, Japan) and accelerated at 3 kV, with a 10 mm working distance and under a vacuum pressure of 40 Pa. Photomicrographs were taken at two magnifications of 350 × and 1000 × at two scales of 100 μm and 50 μm.

### 2.7. Color Measurement

The color of the dry powder of CVLs was measured by using the UltraScan PRO spectrophotometer (HunterLab, Reston, VA, USA). The angle was 45° and the color was expressed in CIE*L*a*b**, where *L**, *a** and *b** denote lightness, redness, and yellowness, respectively. Before testing, standard whiteboard and blackboard calibrations were performed.

The chroma and hue of the CVLs were calculated according to Muhammad et al. [17]:(2)$$C*=\sqrt{{a*}^{2}{+b*}^{2}},$$
(3)$${H{}^{\circ}=\tan h}^{-1}(b*/a*),$$
where *C** represents color saturation and *H*° represents hue.

### 2.8. Extraction of Free Phenolics and Flavonoids from CVLs

A stoppered conical flask was filled with 1 g of the sample (accurate to 0.0001 g), and then 30 mL of 70% (*v*/*v*) ethanol solution was added and extracted three times at 40 ± 2 °C and 350 rpm (average of 40 min per extraction). The solution was diluted to 10 mL with methanol after filtration and concentration and stored in a −20 °C refrigerator away from light [18].

### 2.9. Determination of the Total Polyphenol Content (TPC) and Total Flavonoid Content (TFC)

The TPC of the CVLs was determined by the Folin–Ciocalteu method [18]. Gallic acid was used as the standard for the relative measurement of phenols. In short, the standard solution and sample extract (1 mL) were placed in the calibration colorimetric tube. Folinol (0.5 mL), Na_2_CO_3_ (2 mL, 7.5%, *w*/*w*), and water (6.5 mL) were added. Vortex oscillation was performed in a water bath for 30 min at 70 °C. The absorbance (A) of the standard solution and sample were measured at 750 nm by an ultraviolet spectrophotometer (UV-1206, Shimadzu Corporation, Kyoto, Japan). The measurements were repeated three times, and the results were expressed as gallic acid equivalents per gram (mg GAE/100 g DW).

The TFC was determined by the sodium nitrate–aluminum nitrate method [19]. Rutin was used as a standard for the relative measurement of flavonoid compounds. In short, the standard and sample solutions (1 mL) were placed in the calibration tube, and ethanol solution (4 mL, 30%, *v*/*v*) and NaNO_2_ solution (0.3 mL, 5%, *w*/*w*) were added and allowed to react for 5 min. Then, Al(NO_3_)_3_ (0.3 mL, 10%, *w*/*w*) solution and NaOH solution (2 mL, 1.0 mol/L) were added and allowed to react for 6 min. Finally, 30% (*v*/*v*) ethanol was utilized to dilute the systems. The absorbance (A) of the samples was determined at a 510 nm wavelength. The measurements were repeated three times, and the results were expressed as mg RE/100 g DW per gram of rutin.

### 2.10. Antioxidant Activities

The antioxidant capacities of CLVs were assessed using DPPH, ABTS radical scavenging activities, and ferric reducing capacities, with Trolox serving as a control.

#### 2.10.1. DPPH Radical Scavenging Capacity Assay

The DPPH free radical scavenging assay for all samples was performed according to the method of Cheak et al. [19]. Briefly, extracts (0.2 mL) with different mass concentrations were mixed with DPPH ethanolic solution (3.8 mL, 0.1 mmol/L) and incubated for 30 min in the dark. The absorbance (A) was measured at 517 nm. The extract concentration required to reduce the initial DPPH absorbance by half (IC_50_) was calculated. The formula of the DPPH radical scavenging activity was as follows:(4)$$\mathrm{DPPH}\;\mathrm{radical}\;\mathrm{scavenging}\;\mathrm{activity}\;\left(\%\right)=\frac{A_{0}-\left(A_{1}{-A}_{2}\right)}{A_{0}}\times 100,$$
where A_0_, A_1_, and A_2_ represent the absorbance of ethanol/DPPH solution, different gradient concentration extracts/DPPH, and different gradient extracts/ethanol mixtures at 517 nm, respectively.

#### 2.10.2. ABTS Radical Scavenging Capacity Assay

The ABTS radical solution was prepared by the reaction of the ABTS stock solution (7 mM) and 2.5 mM potassium persulfate. The two solutions were mixed at 1:1 and reacted away from light at room temperature for 12–16 h. The finished combination was then diluted with anhydrous ethanol until it had an absorbance value of 0.7 (±0.02) at 734 nm. In short, samples of different concentrations (1 mL) were mixed with ABTS radical solution (2.5 mL), the absorbance A was measured at 734 nm after 10 min in the dark, and ethanol was used as a blank. The sample concentration that reduced the absorbance of the ABTS radical solution by 50% was calculated and expressed as the IC_50_ value. The calculation formula of the ABTS radical scavenging activity was as follows [19]:(5)$$\mathrm{ABTS}\;\mathrm{radical}\;\mathrm{scavenging}\;\mathrm{activity}\;\left(\%\right)=\frac{A_{0}-\left(A_{1}{-A}_{2}\right)}{A_{0}}\times 100,$$
where A_0_, A_1_, and A_2_ represent the absorbance of ethanol/ABTS solution, different gradient concentration extracts/ABTS, and different gradient extracts/ethanol mixtures at 734 nm, respectively.

#### 2.10.3. Ferric Reducing Capacity Assay

With reference to the methods of Hamrouni-Sellami et al. [20]. some modifications were made to determine the ferric reducing capacity. The samples (0.2 mL), PBS solution (2.5 mL, 0.2 mM, pH of 6.6), and potassium cyanide solution (2.5 mL, 1%, *w*/*w*) were mixed and incubated in a water bath at 50 °C for 20 min. After cooling to room temperature, trichloroacetic acid (2.5 mL, 10%, *w*/*w*) was added. Subsequently, the aliquot solution (2.5 mL), water (2.5 mL), and FeCl_3_·6H_2_O solution (0.5 mL, 0.1%, *w*/*w*) were mixed and reacted for 10 min. The absorbance value (A) was determined at 700 nm. The formula for ferric reducing activity is as follows:(6)$$\mathrm{Ferric}\;\mathrm{reducing}\;\mathrm{activity}\;\left(\mathrm{FRA}\right)=A_{t}{-A}_{0},$$
where A_t_ represents the absorbance at 700 nm of extracts of different concentrations and A_0_ represents the absorbance at 700 nm in the absence of samples.

### 2.11. Statistical Analysis

The results of the experiment are expressed as the means ± standard deviations (SD). OriginPro 2021b (OriginLab Corp., Northampton, MA, USA) was used to create the graphs. GraphPad Prism 9.0 (GraphPad Software, San Diego, CA, USA) was used to analyze the data in this study. The results were considered statistically significant when *p* < 0.05 and 0.01, according to one-way ANOVA. All determinations were repeated 3 times.

## 3. Results and Discussion

### 3.1. Effects of the Drying Method on the Se Content and Species of CVLs

Figure 1A shows the total Se content of fresh CVLs and dried CVLs. The total Se content decreased significantly after drying, showing that the drying procedure induced Se loss, which is consistent with the results of earlier investigations [21]. FIRD samples had the greatest favorable effect on Se retention rate (72.9%), which was 6.26–23.14% higher than those of the other methods. However, FD samples had the lowest total Se retention rate, which was substantially lower than that of the FIRD samples. A possible explanation is that *C. violifolia*, as a super-enriched Se plant, is more prone to losing volatile Se compounds when subjected to a low degree of vacuum during FD. It is also possible that its long-term drying process is also a major cause of Se loss [8]. The VD and MVD data also show that the degree of vacuum has a significant impact on Se and that its content decreases. The specific cause and mechanism of Se loss during drying requires further study. According to Perez et al. [22], the total Se content of garlic decreased by 6–11% after heat treatment, including boiling, steaming, and microwaving. Roasting and frying (180 °C) resulted in a greater loss of Se, demonstrating that too-high temperatures make Se loss more significant. The temperature (60 °C) in this work may not be the main factor for the higher Se loss rate, which may be mainly due to its long drying process and the loss of volatile Se compounds.

In fresh CVLs, SeCys_2_ was the dominant species, accounting for approximately 91.04% of the total Se. SeMet and MeSeCys were detected as two other organic Se species, accounting for approximately 8.48% and 1.34% of the total Se, respectively. Two inorganic Se species, Se(IV) and Se(VI), were discovered to account for 1.92% and 0.69% of the total Se, respectively. Exogenous inorganic Se can be converted to organic Se by plants, which may be related to their ability to metabolize nonprotein selenoamino acids (e.g., Se-methylSeCys and glutamyl-Se-methylSeCys) and nonessential proteins [22]. The influences of different drying techniques on the contents of the five Se species are shown in Figure 1B. The organic Se contents of samples subjected to different drying methods showed significant differences, and the order of organic Se content from large to small was as follows: FIRD, MVD, VD, FD, and HD. The heat stability of different Se species and the effects of different drying processes on Se metabolic pathways could explain this phenomenon. Different drying processes have different effects on the extraction of different active compounds from the same plant; according to a metabolomics study, the freezing or heating of the sample has a significant impact on the metabolic pathways of the extract [23]. This study found that the loss rates of SeCys_2_ and MeSeCys after drying were 15.45–81.63% and 54.06–94.41%, respectively. They have similar structures. Despite the fact that they both feature a heat-sensitive cysteine structure, the thermal stability of the two Se species differs [24], with MeSeCys experiencing greater loss when heated, which is consistent with Perez et al. [22]. Another organic Se species studied, SeMet, had an overall reduction in concentration. The SeMet concentration in the FIRD samples was close to that in the fresh samples. The explanation for this could be that during the FIRD process, the vegetable protein absorbed a large amount of energy, weakening its cohesive effect and making it easier to extract during the enzymatic hydrolysis phase [25].

In CVLs, the effect of the drying technique on inorganic Se was more complicated. Se(VI) changed in a unique way, and the content of Se(VI) in all dried samples was higher than that in fresh samples. High-valent Se(VI) has a higher thermodynamic stability, and other species are more likely to be oxidized to stimulate the creation of Se(VI) under the influence of external factors. Compared with fresh samples, the content of Se(IV) in dried samples generally has different degrees of loss. We found that the content of Se(IV) in MVD samples was slightly higher than that in fresh samples, but the difference was not significant, and Se(IV) was less stable during processing [22]. Among all dried samples, SeCys_2_ was the absolute dominant species, and its content accounted for 74.30–92.08%.

### 3.2. Effects of the Drying Method on Se Bioaccessibility

For in vitro gastrointestinal simulation trials, CVLs with different drying treatments were chosen. Figure 2A–C show the bioaccessibility of total Se and different Se species after simulated gastrointestinal digestion, respectively. The bioaccessibility values of total Se in the G and GI stages were 54.7–73.5% and 59.6–76.6%, respectively. Overall, the bioaccessibility in the GI stage was generally higher than that in the G stage. This result is similar to the bioaccessibility of Se in rice [26], *Pleurotus eryngii,* and chive crops [10,27]. This is because pancreatic enzymes and bile salts in the intestinal juice can degrade polysaccharides and further decompose proteins or polypeptides, thus boosting Se bioaccessibility [26].

The factors affecting the bioaccessibility of Se are mainly determined by the chemical form [3], and numerous studies have demonstrated that Se bioaccessibility is linked to the Se concentration and the type of food consumed [28,29]. Figure 2A shows that the order of total Se bioaccessibility in CVLs after different drying treatments is FIRD > MVD > HD > FD > VD. Similarly to the results of the total Se content after drying, the total Se bioaccessibility of the FIRD samples was the highest, at 76.6%. The second highest were the MVD samples, the bioaccessibility of which was 76.2%. The Se bioaccessibility of VD, FD, and HD samples was low, at 59.6%, 63.7%, and 67.9%, respectively. This may be due to a variation in the microstructure after drying, which is not only related to the overall Se content. The release and use of Se in gastrointestinal fluid is not beneficial for Se bioaccessibility after VD treatment. Another reason for the limited bioaccessibility of Se in VD samples could be the occurrence of Se in *C. violifolia* [7]. During processing, too much of the organic Se content in the selenoproteins and selenopolysaccharides in the samples was lost (as shown in Figure 1B), resulting in low bioaccessibility of Se. However, the mechanism of increased organic Se loss produced by these drying procedures is unknown, and more research is needed.

Figure 2B,C shows the bioaccessibility of different Se species in CVLs subjected to different drying methods. In general, the bioaccessibility of the GI stage is higher than that of the G stage. The bioaccessibility of SeCys_2_ and MeSeCys is similar to the result of total Se, which may be related to their absolute content being the decisive factor affecting their bioaccessibility in the process of gastrointestinal digestion, and according to the statistics, FIRD samples had the highest bioaccessibility, while FD samples had the lowest. The bioaccessibility of SeMet varies dramatically. Organoselenium compounds such as SeMet have been observed to be oxidized during gastrointestinal digestion and extraction, resulting in an oxidized form of SeOMet [4,22]. In this investigation, the improved bioaccessibility of SeMet in the FD sample could be due to its freezing at low temperatures, which renders it more stable and easier to employ in subsequent gastrointestinal simulation trials. The bioaccessibility of Se(IV) and Se(VI) is noteworthy. After gastric digestion, Se(IV) was not detected in FD and VD, and other samples had decreased bioaccessibility. In addition, compared to other species, Se(IV) exhibited the lowest bioaccessibility. Se(VI) and SeCys_2_ showed similar results, and the bioaccessibility of MVD samples reached more than 80%. Studies have shown that Se(VI) is more readily released by enzymatic reactions in gastrointestinal fluid, and its bioaccessibility is relatively high [30].

### 3.3. Morphology of Dried CVLs

All dried CVLs were characterized by SEM to explore the effect of drying on the structure of CVLs. Figure 3 shows the 10 different key images. Based on the different drying methods and drying mechanisms, the samples treated with MVD, FIRD, and HD had obvious shrinkage, which was probably due to the structural shrinkage caused by the long drying time and higher temperature of FIRD and MVD [31]. HD samples had relatively poor drying properties, making the structure significantly wrinkled after drying. The VD-treated samples had an obvious concave cell phenomenon, which had a critical connection with its higher degree of vacuum. The microstructure of FD samples had obvious advantages, and related studies have also shown that FD samples had obvious positive effects on sample structure and cell integrity [32].

Relevant studies have shown that in samples treated with FD, the activities of enzymes such as polyphenol oxidase and peroxide are the highest [33]. The difference is that infrared irradiation can effectively inhibit enzyme activity; lipase and starch are strongly inhibited in these conditions. Polyphenol oxidase is also inhibited to a large extent, causing very little loss of carbohydrates and some water-soluble substances [25], which may lead to the retention of Se. The intact cell morphology of the FD samples and the damaged cell morphology of the FIRD samples may, to some extent, indicate the difference in enzyme activity [23]. This may be an important reason for the significant differences in Se content and Se morphology caused by the different treatments.

### 3.4. Effect of Different Drying Methods on Color

It is critical to maintain the appearance of plants after drying, and Table 1 shows the changes in the color parameters of CVLs under the five drying methods. The color parameter *L** represents the brightness, and the FD and FIRD samples had the maximum and minimum values, respectively, meaning that the color brightness of the FIRD sample was the lowest. Dramatically, this had opposite results to those of the total Se content and bioaccessibility, suggesting that the level of Se did not significantly affect *L**, which was similar to the results of Muhammad et al. [17]. The study found that the diet-fed hens and the control group laid eggs, and the difference in the *L** was not significant (*p* > 0.05); the influence of the brightness value *L** was mainly related to drying conditions, such as time and temperature. FD was used to dry the CVLs at −60 °C, and, thus, it was beneficial to maintain the brightness of the sample, whereas FIRD samples had a lower *L** value due to the higher temperature and longer drying time. The *a** values were all less than zero in the table, suggesting that the dried CVLs were kept green to various degrees, and the *a** value of FD samples was substantially lower than those of the other drying processes, resulting in a more visible green color. The *b** of VD samples was significantly higher than those of other types, indicating that CVLs turned yellow during the drying process. The color angle *H°* was distributed in the range of −71.6–79.8°, which indicates that the CVLs were bluish after drying. In chroma *C**, FIRD samples had the lowest value, indicating that this type had the lowest saturation and the color was relatively dull, while VD had the highest *C** value, indicating that its color was brighter than other types.

### 3.5. Total Polyphenol Content (TPC) and Total Flavonoid Content (TFC) of Dried CVLs

Phenolic compounds and flavonoids are ubiquitous secondary metabolites in plants that have important biological activities and are mainly used to prevent damage caused by oxidative stress [33].

The effect of the drying methods on the TPC of the CVLs is shown in Table 2. Under FD treatment, the polyphenol content was the highest (1134.72 mg/100 g GAE of DW), followed by VD (990 mg/100 g GAE of DW), and HD had the lowest content (514 mg/100 g GAE of DW), with the difference being significant. Intuitively, HD, MVD, and FIRD lead to a reduction in the TPC content, which also indicates that temperature and degree of vacuum play a key role in the content and stability of TPC. In HD samples, the loss of TPC may be related to its drying mechanism. When CVLs dry, water is first removed from the surface, the cells harden, and microstructure damage occurs, which affects the subsequent release of bound phenolics [33].

The trend of the TFC content is significantly different from that of the TPC. The VD samples had the lowest TFC, while the MVD, FD, and HD samples had high TFC (411.01–418.08 mg/100 g RE of DW), and the difference was not significant. The content of flavonoids after FIRD treatment was low, but slightly higher than that of VD. The reason may be that microwaves and higher temperatures are conducive to the cleavage of glycosidic bonds, which releases flavonoids and maintains their content. Furthermore, the loss may be due to the reaction of flavonoids with oxygen during extraction [18].

### 3.6. Assessment of Antioxidant Activities

Table 2 shows the antioxidant activities of the dried CVLs. Among them, FD, VD, and FIRD were the three treatments with the best oxidation resistance, while MVD and HD had the worst oxidation resistance. Table 2 intuitively shows that FD, VD, and FIRD samples had the lowest IC_50_ values; MVD and HD samples had higher IC_50_ values; and FRA showed the same trend. The analysis of the Trolox equivalent antioxidant capacity (TEAC) revealed that the differences in antioxidant capacity between FIRD, VD, and FD were not significant. It is interesting to note that the CVLs, after treatment with these three drying methods, contained different amounts of antioxidant active substances, but were able to show similar overall antioxidant capacity. The reported studies have shown that ice crystals are formed in the plant matrix during the FD process and cause the cell structure to rupture, which is beneficial for polyphenol extraction. The high recovery of the compound enhances the antioxidant activity [18]. The antioxidant capacity of VD samples was also higher, and its higher polyphenol content was the direct reason for its higher antioxidant capacity [33]. Despite its low polyphenol content, the strong antioxidant properties of Se conferred a high antioxidant capacity onto the FIRD samples. MVD and HD have poor antioxidant activity, and the heat generated by them MVD and HD may lead to severe degradation of antioxidant compounds, resulting in changes in polyphenol structure and Se species, in which would lead to lower extraction recovery and low antioxidant capacity. Hence, the antioxidant capacity of CVLs may be directly related to the content of polyphenols and total Se.

### 3.7. Correlation Matrix Analysis

To further explore the effects of polyphenolic compounds, flavonoids, and Se compounds on antioxidant capacity, Pearson correlation coefficient analysis was performed (Table 3). The total Se content (TSC) and TPC were significantly correlated with the DPPH radical, ABTS radical, and FRA (*p <* 0.05). Correlation analysis results showed that polyphenols and Se had strong correlations with DPPH radical, ABTS radical, and FRA, which indicated that these substances have high antioxidant activity levels. The CVLs have a greater total Se concentration, which aids in the scavenging of free radicals in the body. Therefore, it is confirmed that Se, as the active center of Gpx, can both reduce peroxides and participate in the antioxidant process in the body.

## 4. Conclusions

In general, the CVLs prepared by FIRD had the highest total and organic Se retention rates, which were better than those of the CVLs prepared by the other four drying procedures. In terms of bioaccessibility, the FIRD sample group had the most obvious advantage; its total Se bioaccessibility reached 76.6%. It was followed by the MVD group, with a bioaccessibility of 76.2%, and the FD group had the lowest bioaccessibility of Se, at 53.7%. Regarding the bioaccessibility of individual Se species, similar results were found to those for total Se, with the FIRD and MVD groups having the highest bioaccessibility at 80%. Therefore, to minimize the loss of Se and improve the bioaccessibility, it is recommended to use FIRD to treat the CVLs. FD samples had the best microstructure and color, as well as high concentrations of polyphenols and flavonoids, but had low Se content and bioaccessibility. The FIRD samples had the highest Se content, but lower polyphenol content. However, overall, the FIRD and FD samples had high antioxidant activity. This may be the result of the combined effect of Se and polyphenols on antioxidant activity. Thus, further research is needed regarding the health balance of polyphenols and Se content. CVLs treated with FIRD, VD, and FD were shown to have significant antioxidant benefits, while MVD and HD seemed to have the lowest antioxidant effects. Furthermore, the IC_50_ value and TEAC of FIRD also indicated that the samples showed good antioxidant activity levels. These results demonstrate that FIRD exhibits development potential for processing applications in Se supplementation. This study provides new data for the development of appropriate dietary recommendations for adequate selenium intake. Different drying methods cause significant differences in the dietary intake of selenium, and the mechanism of how the drying process affects selenium bioaccessibility remains to be investigated in depth.

## Figures and Tables

**Figure 1 foods-12-00758-f001:**
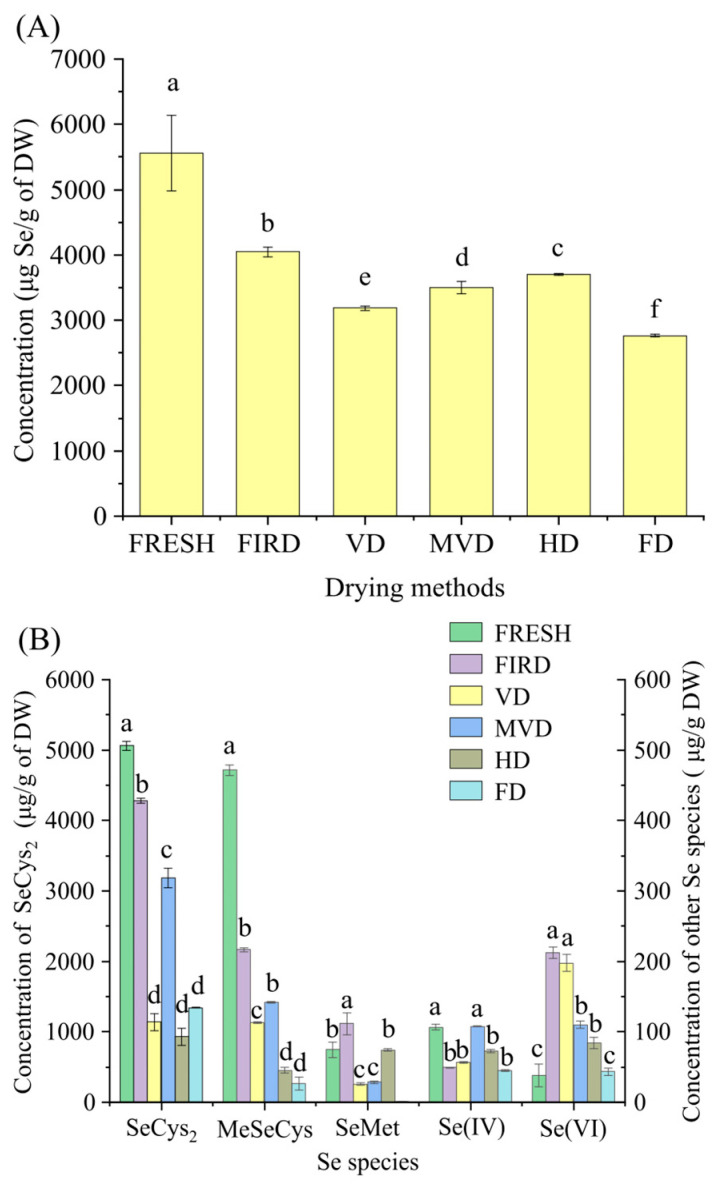
The Se (**A**) and Se species (**B**) concentration of fresh and dried CVLs. Different letters represent significant differences (*p <* 0.05). FRESH: CVLs of fresh samples, FIRD: CVLs of far-infrared drying; VD: CVLs of vacuum drying; MVD: CVLs of microwave vacuum drying; HD: CVLs of hot air drying; FD: CVLs of freeze-drying.

**Figure 2 foods-12-00758-f002:**
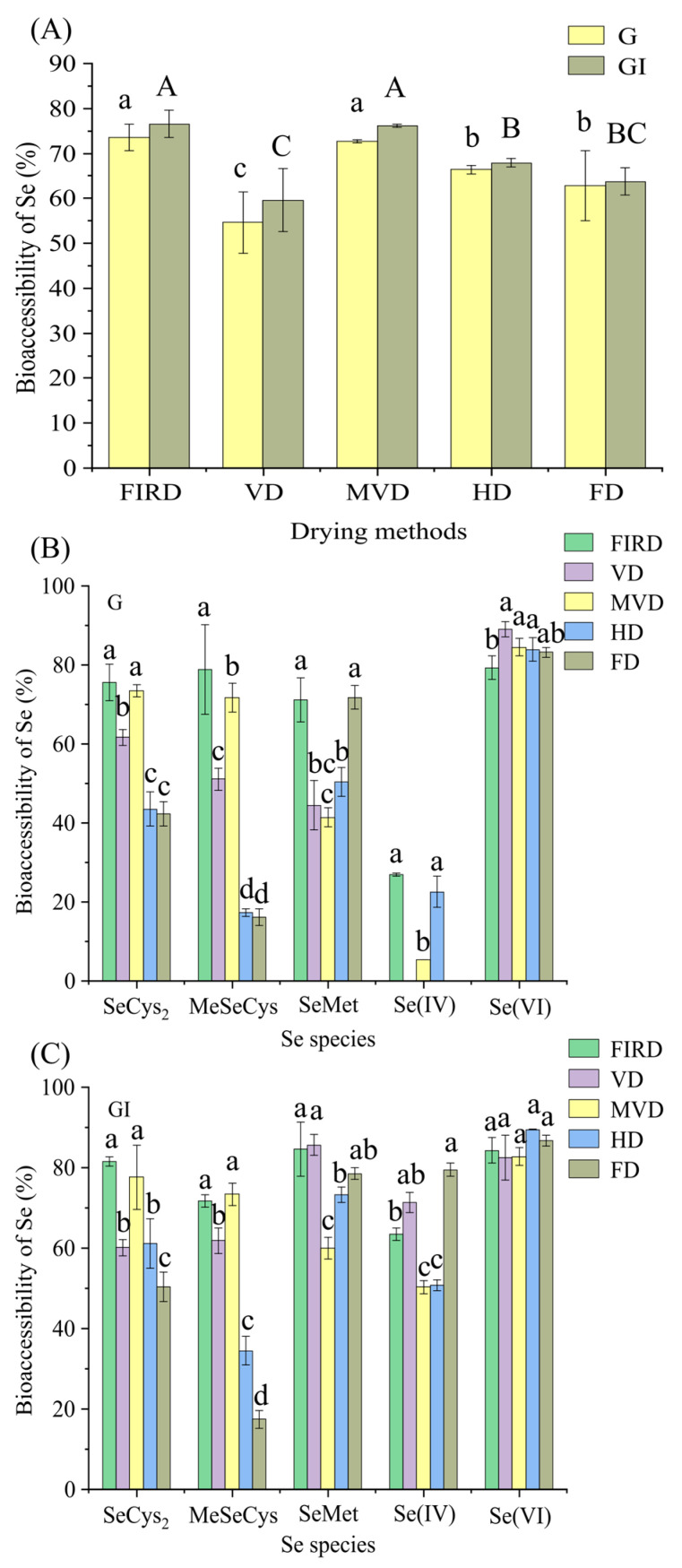
The bioaccessibility of Se (**A**) and Se species (**B**,**C**) in dried CVLs. Different letters represent significant differences (*p* < 0.05). Different Se species groups were subjected to one-way ANOVA in B and C. G: gastric digestion stage; GI: gastrointestinal digestion stage. FIRD: CVLs of far-infrared drying; VD: CVLs of vacuum drying; MVD: CVLs of microwave vacuum drying; HD: CVLs of hot air drying; FD: CVLs of freeze-drying.

**Figure 3 foods-12-00758-f003:**
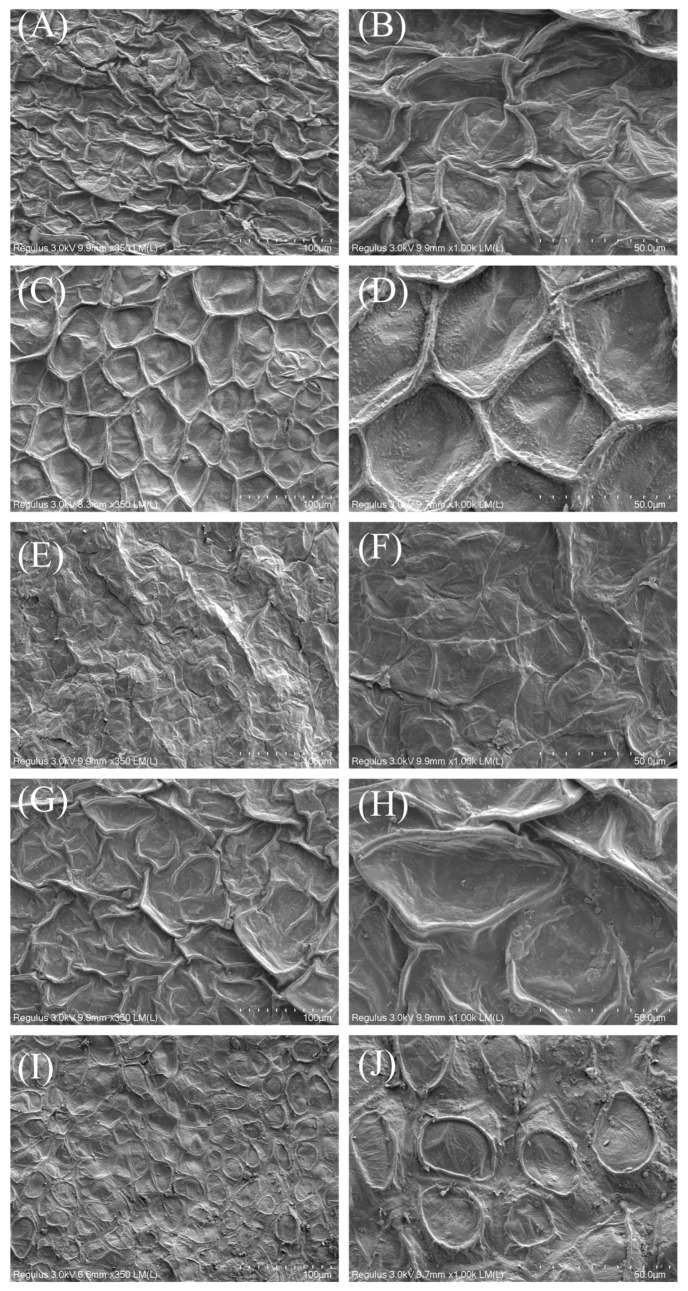
Scanning electron micrographs of (**A**) FIRD, 350×, (**B**) FIRD, 1000×, (**C**) VD, 350×, (**D**) VD, 1000×, (**E**), MVD, 350×, (**F**), MVD, 1000×, (**G**) HD, 350×, (**H**) HD 1000×, (**I**) FD, 350×, and (**J**) FD, 1000× at two different scales (100 μm and 50 μm).

**Table 1 foods-12-00758-t001:** Colors of different dried CVLs.

Drying Methods	*L**	*a**	*b**	*H°*	*C**
FIRD	55.88 ± 0.17 ^e^	−4.50 ± 0.07 ^b^	19.31 ± 0.25 ^e^	−76.88 ± 0.08 ^d^	19.91 ± 0.17 ^e^
VD	60.19 ± 0.21 ^b^	−6.24 ± 0.06 ^d^	24.53 ± 0.35 ^a^	−75.80 ± 0.28 ^c^	25.36 ± 0.30 ^a^
MVD	57.12 ± 0.38 ^d^	−3.95 ± 0.04 ^a^	22.11 ± 0.37 ^bc^	−79.91 ± 0.10 ^e^	22.60 ± 0.11 ^c^
HD	58.43 ± 0.19 ^c^	−6.03 ± 0.06 ^c^	21.19 ± 0.20 ^d^	−74.11 ± 0.12 ^b^	22.04 ± 0.21 ^d^
FD	62.42 ± 0.19 ^a^	−7.26 ± 0.02 ^e^	21.87 ± 0.07 ^c^	−71.60 ± 0.06 ^a^	23.05 ± 0.07 ^b^

Values are means ± SD. The different letters reveal significant differences (*p <* 0.05) according to one-way ANOVA (n = 6).

**Table 2 foods-12-00758-t002:** Total phenolic content, total flavonoid content, and antioxidant activities of dried CVLs.

Drying Methods	TPC (mg GAE/100 g of DW)	TFC (mg RE/100 g of DW)	DPPH· IC_50_ (mg/mL)	ABTS^+^·IC_50_ (mg/mL)	TEAC (μ mol Trolox/g DW)		
					DPPH·	ABTS^+^·	FRA
FIRD	748.18 ± 1.92 ^c^	347.37 ± 6.23 ^b^	22.74 ± 3.98 ^c^	17.77 ± 1.67 ^b^	15.93 ± 1.32 ^a^	25.03 ± 0.52 ^a^	68.39 ± 2.38 ^a^
VD	990.67 ± 9.33 ^b^	320.10 ± 11.15 ^b^	17.19 ± 3.13 ^d^	12.06 ± 0.89 ^c^	17.75 ± 0.78 ^a^	26.20 ± 1.06 ^a^	69.01 ± 3.38 ^a^
MVD	550.81 ± 19.37 ^d^	419.09 ± 17.32 ^a^	36.88 ± 2.97 ^b^	23.07 ± 0.93 ^a^	11.82 ± 2.09 ^b^	22.78 ± 0.52 ^b^	47.26 ± 2.50 ^b^
HD	514.83 ± 17.32 ^d^	411.01 ± 12.45 ^a^	41.12 ± 2.19 ^a^	21.71 ± 0.49 ^ab^	11.21 ± 0.32 ^b^	24.23 ± 0.49 ^b^	42.14 ± 4.75 ^b^
FD	1134.72 ± 9.56 ^a^	418.08 ± 7.95 ^a^	19.62 ± 2.16 ^c^	11.39 ± 2.06 ^c^	16.58 ± 0.91 ^a^	25.76 ± 1.21 ^a^	76.20 ± 2.81 ^a^

The different letters reveal significant differences (*p <* 0.05) according to one-way ANOVA. FRA, ferric reducing activity; TPC, total phenolic content; TFC, total flavonoid content. DW, dry weight; TEAC, trolox equivalent antioxidant capacity of extract concentration at 40 mg/mL; GAE, gallic acid equivalent; RE, rutin equivalent.

**Table 3 foods-12-00758-t003:** Correlation co-efficient between antioxidant activities and TPC, TFC, and TSC in dried CLVs.

	DPPH• Radical Scavenging Activity	ABTS^+^• Radical Scavenging Activity	Ferric Reducing Activity
TPC	−0.9058 *	−0.9775 **	0.9235 *
TFC	0.6437	0.4407	−0.4553
TSC	−0.9756 **	−0.8862 *	0.8843 *

TPC, total phenolic content (mg GAE/100 g of DW); TFC, total flavonoid content (mg RE/100 g of DW); TSC, total Se content (μg/g); * *p <* 0.05; ** *p <* 0.01.

## Data Availability

The data presented in this study are available upon request from the corresponding author. The data are not publicly available due to being used in other future works.

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
