# Peer review of "Effects of Different Drying Methods on the Selenium Bioaccessibility and Antioxidant Activity of Cardamine violifolia"

_foods, 2023, doi:10.3390/foods12040758_

Round 1

Reviewer 1 Report

The manuscript entitles “Effects of different drying methods on the selenium bioaccessi-2 bility and antioxidant activity of Cardamine violifolia” is well structured, and the results about bio-accessibility is interesting for food industry and consumers. Nevertheless, some inconsistencies should be attended.

Please add a justification about the drying conditions. Why were selected?

L100-103, Please improve the dryer equipment description. Lamps, fans, temperature measurement etc. should be mentioning.

L116 . The air velocity in the hot air drying is not mentioned. Please add the information.

L121 Please indicate if the temperature at -60°C is correct.

L180-183 Low pressure, atmospheric, high pressure, which one? also, how was the preparation of the material??, with or without coating? Please improve this section.

L185-190 In Color measurement, Which was the angle of the equipment, 0/45° or other?? Also, is convenient to include total color change.

L291 Se species, Se(IV) and Se(VI), were discovered, This is new in the literature?

L338-339 This is due to a variation in the microstructure after drying, which is not just related to the overall Se content. How the microstructure may influence the bio-accessibility of Se? Please this affirmation is mandatory to improve.

L390-391 The complete morphology of the FD samples and the damage to the cell morphology of the FIRD samples to a certain extent indicate the differences in enzyme activities, which may also be caused. Please add a reference or justify with experimental results.

Table 2 It is not possible to obtains a major concentration of TFC than TPC. TFC are included in TPC.

Reviewer 2 Report

The authors studied the effects of drying on selenium (Se) content, and the bioaccessibility of Se in Se-rich plants is critical. In addition, antioxidant activity was analyzed.

The paper is well structured, however I would have a few comments:

Please complete the abstract with 2-3 more sentences.

In the introduction you wrote nothing about the antioxidant activity of Cardamine violifolia. 

In figure 1 explain what the letters in the columns represent. I assume they represent statistical differences. Why was only figure 1a done and not figure 1b? 

Uniform figures 1 and 2 and make them clearer.

Reviewer 3 Report

THE MS NEEDS  REVISING ACCORDING TO MY COMMENTS THE ILLUSTRATED IN THE ATTACHED PDF COPY. 

Reviewer 4 Report

“Effects of different drying methods on the selenium bioaccessibility and antioxidant activity of Cardamine violifolia”. The manuscript and the research are prepared well. The text is written in scientific language. The study is very interesting because of checking the bioavailability of selenium after various methods of drying the selenium-rich plant, i.e. Cardamine violifolia. There are a few minor errors in the manuscript that need to be corrected:

Line 14 and 78 only here the abbreviation of “microwave vaccum drying” were (MVD); in other place is “WVD” – it must be the same abbreviation

Line 15 and 76-77– should be “Cardamine violifolia” in italic style

Line 16 – “the content of SeCys2 in ….CVLs was the highest” - it was fresh CVLs or after drying?

Line 125-137 should be added some references connected with that method

Line 172 – I think it should be “0.45 µm”

Line 280 – without “a”

Line 312-320 – there is no references in that part of manuscript

Figure 2 Have been done bioavailability analyzes for the fresh plant?; the graphs should be in the same size.

Line 467 – here should be expansion of abbreviation TSC

Line 555 - there is no reference no 19 in the text. Hamrouni-Sellami reference in the text has got number 20.

Reviewer 5 Report

It is a very interesting paper, methodology is well covered, results are very interesting, nevertheless some changes could improve the manuscript.

Line 16; the best drying method was FIRD for the organic Se contain, however authors write “The content of SeCys2 in CVLs was the highest (5060.50 μg/g of DW). After FIRD, the organic Se loss was less than 19%”, as it was redacted is not clear that FIRD was the best method.

FIRD was the best method to retain organic Se in the different forms, nevertheless a decrease in flavonoids and total phenolics was observed compared with the other drying methods, Se is important for health but phenolics too, can the authors discuss more about this balance?.

Reviewer 6 Report

The study “Effects of different drying methods on the selenium bio accessibility and antioxidant activity of Cardamine violifis quite interesting but required further improvement.

Title: title should reflect the findings rather a general idea and plan of study “X/Y/Z method enhances the bio accessibility and antioxidant activity of Cardamine violif”

Abstract

Line 12: Define Se firstly and use abbreviation later on. Same on other places in ms. E.g. FIRD etc.

Line 13: Se-bio accessibility in Se-rich plants is critical. Why critical? Explain it.

Line 19: Add some numeric figs of results rather stating generally in the abstract.

Introduction

1.      Describe some critical limits/values of Se toxicity

2.      Information about the factors especially drying methods on the se bioavailability is also lacking.

3.      Please fill the gap between the set objectives and earlier reports

Materials and methods

Overall, each segment should be described in detail. Current information is not sufficient for reproducibility of the experiments. Concentration of some ingredients need to be mentioned as final conc in the reaction rather describing their stock conc.

Results & Discussions

Footnotes of figs should be revised. E.g . I could not see the description of error bars in footnotes of Fig1&2. Same comment for tables. What does + - signs represent. Please check all figs/Tables for such necessary description. The results need to be discussed logically by adding the proper reasoning.

 References

Please check the references critically for consistency and accordance to the journal’s instructions.

Round 2

Reviewer 1 Report

The authors attend all recommendations

Reviewer 3 Report

the MS is ok now